# Infection Levels of the Microsporidium *Larssoniella duplicati* in Populations of the Invasive Bark Beetle *Ips duplicatus*: From Native to New Outbreak Areas

**Soňa Zimová [1,\*], Karolina Resnerová [1]** , **Hana Vanická [1], Jakub Horák [1]** , **Jiří Trombik [1],**
**Magdalena Kacprzyk [2]** , **Åke Lindelöw [3], Mihai-Leonard Duduman [4]** **and Jaroslav Holuša [1]**

[1] Department of Forest Protection and Entomology, Faculty of Forestry and Wood Sciences, Czech University of Life Sciences Prague, Kamýcká 129, Prague 6-Suchdol, CZ-16500 Prague, Czech Republic; resnerovak@fld.czu.cz (K.R.); vanicka@fld.czu.cz (H.V.); jakub.sruby@gmail.com (J.H.); trombik@fld.czu.cz (J.T.); holusa@fld.czu.cz (J.H.)

[2] Department of Forest Protection, Entomology and Forest Climatology, Institute of Forest Ecosystem Protection, University of Agriculture in Krakow, Aleja Adama Mickiewicza 21, PL 31-120 Kraków, Poland; magdalena.kacprzyk@urk.edu.pl

[3] Department of Ecology, Forest Entomology Unit, Swedish University of Agricultural Sciences, SLU Department of Ecology, Box 7044, SE-75007 Uppsala, Sweden; ake.lindelow@slu.se

[4] Applied Ecology Laboratory, Faculty of Forestry, Ștefan cel Mare University of Suceava, Str. Universitatii 13, RO-720229 Suceava, Romania; mduduman@usv.ro

\* Correspondence: zimovas@fld.czu.cz; Tel.: +420-731-069-242

**Abstract:** The microsporidium *Larssoniella duplicati* (Weiser, Holuša, Žižka, 2006) is a specific pathogen of the bark beetle *Ips duplicatus* (C.R. Sahlberg, 1836), which is a serious pest of Norway spruce (*Picea abies* (L.) H. Karst) in Europe. From 2011 to 2016, infection levels of *L. duplicati* and other pathogens in *I. duplicatus* populations were assessed along a gradient, ranging from areas in the north, where the beetle is native, to areas in the south, where the beetle has only recently invaded. The 21 study sites ranged in altitude from 229 to 1009 m a.s.l. We found that pathogen infection levels in *I. duplicatus* populations decreased from the native areas in the north to the new areas of beetle expansion in the south. We also found that pathogen level increased with altitude. The *L. duplicati* infection levels were not associated with the infection levels of other beetle natural enemies. The infection level decreased with the length of time of beetle establishment in an area. The infection level increased with the number of beetles trapped and dissected at a site.

**Keywords:** *Ips duplicatus*; pathogen; vector; infection level; invasion; latitude

## 1. Introduction

Changes in climate and land use can increase the spread of organisms [1]. Many of these organisms are non-native to their new area of distribution; some spread to new areas but also increase their population densities in their former areas [2,3]. In some cases, such invasive species begin to damage habitats that are important for humans, like forests with fast-growing tree species [4]. One of the most commercially important tree species in Europe is the Norway spruce (*Picea abies* [5]). This tree is attacked by many species of bark beetles of which *Ips typographus* (Linnaeus, 1758) is the most important in terms of loss of mature trees before final cutting [6].

The double-spined spruce bark beetle *Ips duplicatus* is a native species in Scandinavia, eastern and northern parts of central Europe and northeast Asia, where it occurs on Norway spruce. The beetle is currently spreading to Norway spruce in many parts of Europe. Its high outbreak potential is

supported by climatic change, the physiological weakness of trees, and the attack of such weakened trees by the fungus *Armillaria ostoyae* (Romagn. Herink, 1973) and other pathogens [7,8]. Current studies focusing on wind and bark beetle disturbances suggest increase damages in Europe under climate change [9–11]. The combination of increasing frequency of drought events, Norway spruce planting in non-native habitats and warmer temperatures are considered important predisposing factors triggering the double-spined spruce bark beetle outbreaks. As a result of these factors affected by climate change, the number of *Ips duplicatus* generations is increasing to two to three during one vegetation period in the central European area [12].

From the beginning of the 20th century, the beetle began spreading from its origin in the Palearctic region to the south because spruce monocultures were being increasingly established in the south in Europe [8], unlike most other bark beetle invasions that extend from south to north [13,14]. *I. duplicatus* was first noted in eastern Czech Republic and south Poland in 1960s [15–17]. That area experienced massive *I. duplicatus* outbreaks in the 1990s. During the last 200 years, Norway spruce has been planted in many areas of Europe, mainly out of the natural range of this tree. As the planted trees are growing out of their natural range, they may be stressed [18], and this has increased the spread of *I. duplicatus* to southern Europe [19]; *I. duplicatus* has even been recorded in south Slovakia [15,20] and throughout Romania [21].

The microsporidium *Larssoniella duplicati* appears to be a specific pathogen of *I. duplicatus*; its presence in other spruce bark beetles, such as *I. typographus*, *Pityogenes chalcographus* (Linnaeus, 1761), and *Ips amitinus* (Einchoff, 1871), has not been reported [22–24]. This specificity of *L. duplicati* is not as usual among pathogens of bark beetles; i.e., the same pathogen usually occurs in multiple bark beetle species, but other examples are known [25–29].

*L. duplicati* was first described in the Czech Republic and Poland [24], where its infection levels in *I. duplicatus* populations are stable and where the disease is probably chronic [22]. This microsporidium infects the midgut muscularis, the ovaries, and the Malpighian tubules of adult beetles. The infection is always in the infected tissue, because infected muscle fibres hold the spores in position [24]. Its infection levels in the native area of beetle (Scandinavia) and the new outbreak area (Romania) have not been studied [23].

The current study had two objectives. The first was to compare the infection levels of *L. duplicati* in the native and new outbreak areas of *I. duplicatus* in Europe. The second objective was to identify variables associated with differences in *L. duplicati* infection levels in *I. duplicatus* populations.

## 2. Materials and Methods

Pathogens of *I. duplicatus* were studied at 21 sites: four in the Czech Republic, five in Romania, eight in Poland, and four in Sweden. The altitudes of study sites ranged from 229 to 1009 m a.s.l. (Figure 1). During the years of 2011–2016, beetles were collected using Theysohn pheromone traps (Theyson Kunststoff. GmbH, Germany) or Intercept traps (only in Romania) baited with pheromone lures ID Ecolure (FYTOFARM Group s.r.o., Slovakia), Pheagr IDU (Sci-Tech, s.r.o, Czech Republic), Duplodor (Chemipan, Poland), or an experimental lure (Romania) [30] (Table 1). In all used pheromone lures, the main compound is always E-myrcenol—the main aggregation pheromone component for *I. duplicatus* [31]. The pheromone lures were changed after 10 weeks. Each site was sampled in only 1 or 2 years.

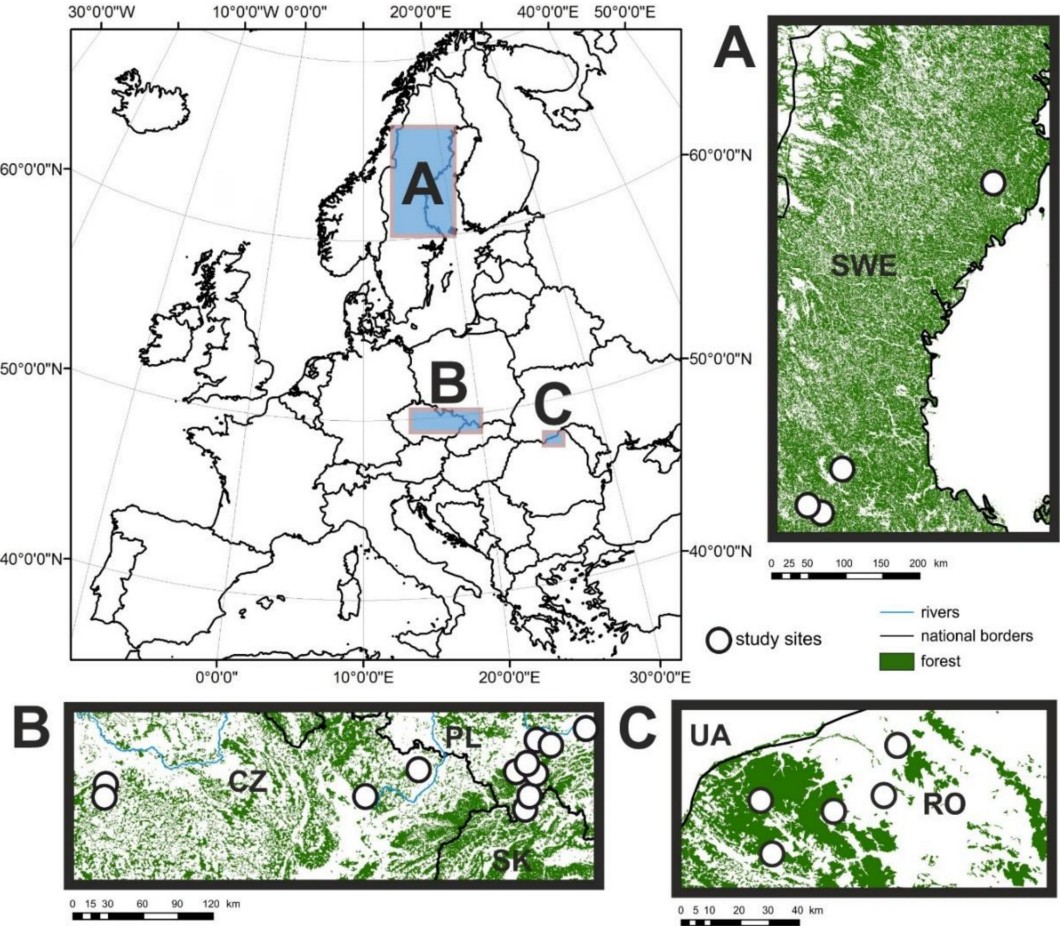

**Figure 1.** Study sites (circles) in Europe where *Ips duplicatus* was collected during 2011–2015 in forest areas (green).

Beetles were collected from the beginning of May to the end of August. In the study sites, flight barrier traps were placed 1.5 m above the ground and approximately 15–20 m from a standing spruce tree that was more than 30 years old. All forest stands at study sites were composed of a mosaic of trees of all ages, so there was enough suitable material for *Ips duplicatus* infestation.

Trapped beetles were placed in Eppendorf micro-test tubes with a piece of damp gauze to maintain humidity. The tubes were stored frozen until the beetles were dissected.

Each beetle was identified to species [32] and then dissected by removing the gut, Malpighian tubules, ovaries, and the body fat. The dissected tissues were examined with a light microscope (Nikon Eclipse 50 Ni, Nikon Instruments Inc., Melville, NY, USA) at 40 to 400× magnification to determine the presence of *L. duplicati* (oval spores of two sizes, 3–3.5 × 1.5–2 and 2–2.5 × 1.5 in intestinal muscles) and other pathogens and nematodes.

Data concerning the distribution of coniferous forests relative to the study sites were obtained from [32] and were corrected using Corine Land Cover. The program ArcMap 10.0 (ESRI, Redlands, CA, USA) was used to create Figure 1, which shows the distribution of the study sites.

Basic statistical analyses were performed in Statistica 13.1 (Dell software, Austin, TX, USA). We used the Shapiro Wilk test to determine the normality of the data (infection level). The Wilcoxon matched pair test was used to compare infection levels between sexes (percentages of infected males vs. females). Non-parametrical analyses were used as a control for the potential influence of local differences in infection levels at the country level.

Detailed analyses were done in SAM v4.0 [33]; we computed Moran's I to assess the spatial autocorrelation of our dependent variable (infection level of *L. duplicati*) in seven distance classes.

We assessed the relationships between infection level of *L. duplicati* (the percentage of infected individuals at a site, and the dependent variable) and the following independent variables: altitude, latitude (north-south gradient), longitude (east-west gradient), infection level of *Chytridiopsis typographi* parasitism by intestinal nematodes, parasitism by hemolymph nematodes, number of *I. duplicatus* beetles captured and dissected, and year (time of beetle collection). For linear regression of infection level on independent variables, infection level data were arcsine square root transformed to obtain normality. Analyses of the interaction among studied independent variables indicated multicollinearity for longitude (VIF = the variance inflation factor >2), which was the variable that described the east-west gradient in outbreak area of *I. duplicatus*. Thus, longitude was not further analysed. As some independent variables were not significant, we selected variables for inclusion in the final model based on AICc (Akaike information criterion with correction for small sample sizes) as implemented in SAM. In further analyses of *L. duplicati*, we used seven independent variables: latitude (north-south gradient); the infection level of the microsporidium *Chytridiopsis typographi*; the parasitism by intestinal nematodes; the parasitism by nematodes in hemolymph (hereafter termed hemolymph nematodes); the altitude of the study site; the number of *I. duplicatus* beetles trapped and dissected at a site; and the year of beetle collection.

**Table 1.** Background information on the study sites where *I. duplicatus* specimens were collected and assessed for pathogen infection. Country of origin (Country): Sweden (SWE), Poland (PL), Czech Republic (CZ), Romania (RO). In traps were used different pheromone lures: ID Ecolure, Duplodor, Pheagr IDU and in Romania the experimental lure (exp. lure) [30].

| Study Sites | Country | GPS Coordinates | | Year of Collection | Pheromone Lure | Altitude (m a.s.l.) |
|---|---|---|---|---|---|---|
| | | N | E | | | |
| Nås | SWE | 60.4677 | 14.5003 | 2014 | ID Ecolure | 232 |
| Siljansfors | SWE | 60.9730 | 15.0578 | 2014 | ID Ecolure | 324 |
| Vansbro | SWE | 60.5229 | 14.2389 | 2014 | ID Ecolure | 229 |
| Vindeln | SWE | 64.2000 | 19.7833 | 2014 | ID Ecolure | 291 |
| Petkówka | PL | 49.7333 | 19.2333 | 2015; 2016 | Duplodor | 668 |
| Rajcza | PL | 49.7666 | 19.2333 | 2015; 2016 | Duplodor | 646 |
| Romanka Górna I | PL | 49.5805 | 19.2246 | 2016 | Duplodor | 829 |
| Romanka Górna II | PL | 49.9338 | 19.3989 | 2015 | ID Ecolure | 1009 |
| Sopotnia Dolna | PL | 49.9350 | 19.4664 | 2015 | ID Ecolure | 953 |
| Tokarnia | PL | 49.9833 | 19.9833 | 2015 | ID Ecolure | 688 |
| Ujsoły | PL | 49.7508 | 19.2009 | 2015; 2016 | Duplodor | 859 |
| Złatna | PL | 49.4833 | 19.1666 | 2015 | ID Ecolure | 638 |
| Hlubočky | CZ | 49.6920 | 17.4146 | 2013 | ID Ecolure | 382 |
| Jílové u Prahy I | CZ | 49.8866 | 14.5055 | 2016 | Pheagr IDU | 354 |
| Jílové u Prahy II | CZ | 49.9166 | 14.5071 | 2016 | Pheagr IDU | 457 |
| Pustá Polom | CZ | 49.8510 | 18.0242 | 2014 | ID Ecolure | 454 |
| Calafindești | RO | 47.8513 | 26.1459 | 2011 | exp. lure | 497 |
| Ionu | RO | 47.6134 | 25.4817 | 2013 | exp. lure | 1080 |
| Solca | RO | 47.7000 | 25.7963 | 2013 | exp. lure | 625 |
| Sucevița | RO | 47.7767 | 25.4817 | 2013 | exp. lure | 605 |
| Todirești | RO | 47.7127 | 26.0328 | 2013 | exp. lure | 415 |

## 3. Results

A total of 1539 adults of *I. duplicatus* from the 21 study sites located throughout the Czech Republic, Romania, Poland, and Sweden were dissected and analyzed.

The *L. duplicati* infection level in *I. duplicatus* populations (i.e., the percentage of specimens at a site with *L. duplicati*) across all countries averaged ± standard error (SE) 16.7% ± 8.4% and ranged from 0% to 39.1%. *L. duplicati* was detected in 20 of the 21 sites (Table 2). *L. duplicati* infection levels did not significantly differ between *I. duplicatus* sexes (Z = 1.33, *p* > 0.05). Infection occurred only in the intestinal muscles of *I. duplicatus*.

**Table 2.** Infection levels of four pathogens in *I. duplicatus*. Infection level refers to the percentage of beetles with the indicated pathogen. The location of the study site (Country): Sweden (SWE), Poland (PL), Czech Republic (CZ), Romania (RO). For each study site there is a number of inspected beetles (N) and infection levels of: *Larssoniella duplicati* (*L.d.*), *Chytridiopsis typographi* (*C.t.*), parasitism by intestinal nematodes (I.n.) and hemolymph nematodes (H.n.).

| Study Sites | Country | N | *L.d.* (%) | *C.t.* (%) | I.n. (%) | H.n. (%) |
|---|---|---|---|---|---|---|
| Nås | SWE | 46 | 39.1 | - | 15.2 | - |
| Siljansfors | SWE | 70 | 21.4 | 1.43 | 10.0 | 4.3 |
| Vansbro | SWE | 156 | 16.7 | - | 3.2 | 1.3 |
| Vindeln | SWE | 72 | 23.6 | - | 11.1 | 5.6 |
| Petkówka | PL | 107 | 19.6 | - | 3.8 | 4.6 |
| Rajcza | PL | 103 | 13.6 | - | 14.1 | 5.5 |
| Romanka Górna I | PL | 27 | 7.4 | - | 14.8 | - |
| Romanka Górna II | PL | 192 | 20.8 | - | 4.7 | 7.3 |
| Sopotnia Dolna | PL | 35 | 25.7 | - | 5.7 | 2.9 |
| Tokarnia | PL | 139 | 19.4 | - | 6.5 | 3.6 |
| Ujsoły | PL | 22 | 9.1 | - | 13.6 | 9.1 |
| Złatna | PL | 20 | 10.0 | - | 10.0 | - |
| Hlubočky | CZ | 22 | 13.6 | - | 18.2 | 4.6 |
| Jílové u Prahy I | CZ | 18 | - | - | 5.6 | - |
| Jílové u Prahy II | CZ | 43 | 7.0 | 2.3 | 4.7 | 4.7 |
| Pustá Polom | CZ | 237 | 27.4 | 0.8 | 10.1 | 1.7 |
| Calafindeşti | RO | 20 | 20.0 | - | 10.0 | - |
| Ionu | RO | 33 | 18.2 | - | 12.1 | 3.0 |
| Solca | RO | 80 | 11.3 | - | 3.8 | 3.8 |
| Suceviţa | RO | 45 | 8.9 | - | 13.3 | 6.7 |
| Todireşti | RO | 52 | 1.9 | - | 5.8 | 9.6 |

Average levels of *L. duplicati* infection did not significantly differ among countries (H = 4.96; *p* > 0.05). The *L. duplicati* infection level increased from south to north, averaging 12.1% ± 6.5% in Romania, 15.7% ± 6.1% in Poland, 16.1% ± 8.5% in the Czech Republic, and 25.2% ± 8.4% in Sweden (Table 2).

The microsporidium *Chytridiopsis typographi* ((Weiser, 1954) Weiser, 1970) was found at only three study sites, and these were in the Czech Republic and Sweden. Its infection levels were very low (Table 2).

In contrast, nematodes were found in *I. duplicatus* at 21 study sites. The parasitism rate ranged from 3% to 16% for intestinal nematodes and from 0% to 10% for hemolymph nematodes (Table 2). For both kinds of nematodes, average parasitism rate did not significantly differ among countries (intestinal nematodes: H = 0.08; *p* > 0.05; nematodes in the hemolymph: H = 0.81; *p* > 0.05).

The spatial autocorrelation for *L. duplicati* infection levels was not significant (Table 3). This indicated that infection levels tended to be randomly distributed in space, without a tendency toward clustering or regular spacing. The expected Morans I value was −0.06.

**Table 3.** Statistics for spatial autocorrelation analysis of *L. duplicati* infection levels in *I. duplicatus* populations at the 21 sites in Europe.

| Distance Class | Distance Centre | Moran's I | *p* |
|---|---|---|---|
| 1 | 45.2 | 0.1 | 0.6 |
| 2 | 306.6 | 0.1 | 0.7 |
| 3 | 650.3 | −0.2 | 0.2 |
| 4 | 877.8 | −0.1 | 0.6 |
| 5 | 1137.8 | 0.1 | 0.7 |
| 6 | 1510.1 | −0.1 | 0.9 |
| 7 | 1975.8 | −0.4 | 0.1 |

In regression analyses, the *L. duplicati* infection level was significantly related to altitude, latitude, year, numbers of dissected beetles, and the infection level of all other pathogens (F = 6.63; $p < 0.01$; Table 4). A regression model with all of the variables listed in Table 4 (significant and non-significant) explained a total of 71.2% of the adjusted variance in the *L. duplicati* infection level. The *L. duplicati* infection level was not significantly related with the infection levels of *C. typographi*, parasitism by intestinal nematodes, or hemolymph nematodes. The *L. duplicati* infection level significantly increased with latitude, altitude, and the number of beetles captured and dissected at a site, but significantly decreased with the year of the study (Table 4).

**Table 4.** Results for a regression model describing the relationship between the *L. duplicati* infection levels in *I. duplicatus* populations and the following variables: latitude (north-south gradient); infection level of *C. typographi*; parasitism by intestinal nematodes; parasitism by hemolymph nematodes (i.e., nematodes detected in the hemolymph); altitude; number of *I. duplicatus* beetles captured and dissected; and year (date of beetle collection). Variance Inflation Factor (VIF), corrected Akaike's Information Criterion (AICc) = −11.93. Significant variables are in bold.

| Variable | VIF | *t* Value [a] | *p* Value |
|---|---|---|---|
| **Constant** | | **3.1** | **0.01** |
| **Latitude** | **1.1** | **3.5** | **0.01** |
| *C. typographi* | 1.3 | 1.4 | 0.18 |
| Intestinal nematodes | 1.4 | 0.8 | 0.43 |
| Nematodes in hemolymph | 1.1 | −0.8 | 0.46 |
| **altitude** | **1.4** | **3.8** | **0.01** |
| **number** | **1.1** | **2.9** | **0.02** |
| **year** | **1.6** | **−3.4** | **0.01** |

[a] Positive and negative *t* values indicate positive and negative associations, respectively.

In the next step of the statistical analysis, we deleted non-significant variables from the model; the resulting model explained 70.1% of the adjusted variance in ($r^2$adj = 0.701; Table 5). We found that only significant variables from the previous regression left in the model and their *p* values were more significant, except of number of dissected beetles.

**Table 5.** Results of the model that best described (delta AICc <2 based) the relationship between the *L. duplicati* infection level in *I. duplicatus* populations (arcsine square root transformed). The best model included four predictor variables: latitude (north-south gradient); altitude; number of *I. duplicatus* beetles captured and dissected; year (date of beetle collection). Variance inflation factor (VIF). Corrected Akaike's Information Criterion (AICc) = −25.95 (significant variables are in bold).

| Variable | VIF | *t* Value | *p* Value |
|---|---|---|---|
| **Constant** | | **3.9** | **0.002** |
| **Latitude** | **1.0** | **4.0** | **0.002** |
| **Altitude** | **1.2** | **3.7** | **0.002** |
| **Number** | **1.0** | **2.8** | **0.020** |
| **Year** | **1.1** | **−3.9** | **0.002** |

## 4. Discussion

The current research studied the species-specific pathogen *L. duplicati* associated with the double-spined spruce bark-beetle in areas where *I. duplicatus* is native, as well as in areas where *I. duplicatus* is newly established in Europe. We found two interesting patterns: *L. duplicati* infection levels in *I. duplicatus* populations significantly decreased across the latitudinal gradient from the north to the south and significantly increased with increasing altitude.

In the areas where *I. duplicatus* is native, the *L. duplicati* infection level was as high as 30%; in the areas experiencing new outbreaks of the beetle, infection levels varied around 10% [34]. In the current

study, the highest infection level was 39.1%, and the infection level was higher than 10% at most of the study sites, what is consistent with previous reports [22–24]. The spatial distribution of infection levels was not influenced by the spatial arrangement of the study sites (i.e., sites with high or low infection levels did not tend to cluster in space). This was true even though some of the sites, especially those in Poland and the Czech Republic, were located near areas with spruce forests that have been highly stressed by drought and fungal diseases. Such stressed forests typically support higher population densities of *I. duplicatus* than non-stressed forests [20,35]. Generally, latitude-altitude gradient can be explained by increasing number of individuals in population at northern study sites and in long-term outbreak areas. Study sites with more abundant populations of bark beetles are collected more often and with higher infection levels of pathogens [36].

We also suspect that *L. duplicati* may influence the invasive potential and spread of *I. duplicatus*. This is because *L. duplicati* is likely to reduce the fitness of the infected beetles and infection level is growing more slowly in the newly established outbreak areas. In addition to being infected by *L. duplicati*, *I. typographus* and related bark beetles are also attacked by ectoparasitoids and by the pathogen *Mattesia schwenkei* Purrini, 1977. Infection level of this antagonists of *I. typographus* had lower infection levels in areas with new outbreaks of the beetle than in areas with long-lasting outbreaks (more than 10 years) of the beetle [37]. When bark beetle numbers are low or when contacts between individuals in breeding systems are limited, e.g., as is the case in managed forests, there is a reduced probability of pathogen transmission and therefore a low infection level of some common pathogens [23].

The infection level of *L. duplicati* increased with the number of individuals dissected at a site. Nevertheless, the infection levels do not change with changes in host population density [22–24,34], which suggests that transmission is vertical rather than horizontal [23], as it is for some other microsporidium pathogens [29,38]. Therefore, it is unclear why the infection level should increase with number of analyzed beetles of *I. duplicatus*. In the case of horizontally transmitted pathogens, the infection levels sometimes double or triple during the beetle reproductive period of even one generation [39]. In the current study, the main factor associated with low *L. duplicati* infection levels in *I. duplicatus* was the length of time that the area had been infested with the beetle. This effect of time since beetle establishment is somewhat unclear in the current study, however, the latter factor was confounded with collection date.

*I. duplicatus* produces only one generation per year in the boreal forests and in northern Poland [8,40] but up to three generations per year in Central Europe [12,15,41]. Although new outbreaks of *I. duplicatus* occur only sporadically at higher altitudes [16,20,42–44], the *L. duplicati* infection level was related to altitude in the current study. This could be explained by the relationship between latitude and altitude, i.e., the more southern sites had both low infection levels and low altitudes.

We also found that the *L. duplicati* infection level did not differ between *I. duplicatus* sexes or among the studied countries, which is consistent with previous reports for *L. duplicati* as well as for other pathogens of bark beetles [39,45].

The only insignificant relationships between *L. duplicati* infection levels and the other variables were with the infection levels of *C. typographi*, parasitism by intestinal nematodes, and hemolymph nematodes. Nematodes and *C. typographi* are the most frequently reported antagonists of *I. duplicatus* [23,26,27,46]. The infection level of *C. typographi* is often very low [23]. In our study, we found *C. typographi* at only three sites, and infection was always less than 2.4%, suggesting that *C. typographi* was probably not affecting *I. duplicatus* population density. Parasitic nematodes are commonly associated with *I. duplicatus*, occurring in more than 70% of the beetle's gallery systems [46–49]. As in the case of *C. typographi*, nematodes did not appear to affect *L. duplicati* infection levels.

## 5. Conclusions

*L. duplicati* is probably a chronic pathogen of *I. duplicatus* and might have little or even no negative effect on the beetle—especially out of the native distribution area of its host. This microsporidium may negatively influence the flight capability of pioneer beetles and their ability to successfully invade new host trees, but in time (few years) the infection level of this microsporidium increases in a new population and the differences are minimized. The infection levels of *L. duplicati* in *I. duplicatus* populations decreased with latitude; it was highest in the north (Sweden), where the beetle is native, and was lowest in the south (Romania), where the beetle has only recently invaded. This is most probably connected with colonization aspect of a new sites. Infection levels increased with altitude, but the effect of altitude was confounded with the effect of latitude.

The most important conclusions of our research on an alien pest and its pathogen is that they follow a latitude-altitude gradient. This most probably reflects fact that spread of pathogen is prolonged (e.g., similar to known escape from enemies' hypothesis in bark beetles) [50]. Nevertheless, altitude in coincidence with latitude, indicate some climatic limits of the pathogen—as north sites and high elevations are often more cold and wet than the opposite. This is also important regarding pest management. Even if *L. duplicati* does not have a strong impact on alien bark beetle, its virulence could have some impact on invasive success of the bark beetle.

**Author Contributions:** S.Z., K.R., and H.V. provided methodology of research and collected field or laboratory data; J.H. (Jakub Horák) and J.T. performed the statistical analyses, and the visualizations; M.K., A.L., and M.-L.D. managed the research and collection of material in their country and took part in finalizing the manuscript; K.R., J.H. (Jakub Horák) and J.H. (Jaroslav Holuša) supervised the research and edited the manuscript. All authors provided feedback on the manuscript and approved the final manuscript.

**Funding:** This work was supported by the Czech University of Life Sciences Prague project No. IGA C_01_18, by grant "Advanced research supporting the forestry and wood-processing sector's adaptation to global change and the 4th industrial revolution", No. CZ.02.1.01/0.0/0.0/16_019/0000803 financed by OP RDE and by grant No. QK1920433 of the Ministry of Agriculture of the Czech Republic.

**Acknowledgments:** The authors thank Bruce Jaffee (USA) for linguistic and editorial improvements.

**Conflicts of Interest:** The authors declare no conflict of interest.

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
