# Peer review of "Infection Levels of the Microsporidium Larssoniella duplicati in Populations of the Invasive Bark Beetle Ips duplicatus: From Native to New Outbreak Areas"

_forests, doi:10.3390/f10020131_

Round 1
Reviewer 1 Report
Generally, I miss information on infected tissue, this has some importance because the authors discuss about the horizontal and vertical transmission of Larssoniella duplicati. If such data exist it would be necessary to give it in the paper if not a statement about this would be appreciating.
Line 44 can you please provide more information about climate change, because usually climate change in context of pest spreading is from south to north (many references about that)
Lines 56-60 I think there are some information missing considering the specificity of L.t. infection on just one bark beetle species. There is evidence when pathogen occur only in a genus of bark beetle species. This is not the same as Chytridiopsis typographi which is present in many genera. I think this is worth to mention because the are some examples in the literature (Caningia in Tomicus or Pityokteines) and C.t. is also a part of the presented study.
Lines 80-82 Why did you change the sites of traps from year to year (more or less)? Data would be more clear if just the same site would appear every year
Line 87 Can you provide exact how old was a specific forest on specific site? This possible would have an impact on the abundance and possible also on the infection level. This should be clarified
Line 140 It looks correct mathematically but are the data presented for CZ really correct' Because there is a high infection rate in just one site (?). Please give an explanation for that.
Line 221-228 in line 208 you stated that infection of L.d. do not change with host population density but in this line it becomes unclear. Maybe you should rewrite these two lines making it more clear. They are some references where population of bark beetle in outbreaks an impact on infection levels of a pathogen have (eg Canningia Pityokteines) which should be mentioned too.
Author Response
Generally, I miss information on infected tissue, this has some importance because the authors discuss about the horizontal and vertical transmission of Larssoniella duplicati. If such data exist it would be necessary to give it in the paper if not a statement about this would be appreciating.
- The lines 68-71 describes where the infection in the beetle body is localized and we add information about spores in muscle fibres. We think it is not necessary to write more about that, because all that information is in the literature cited there: Weiser, J.; Holuša, J.; Žižka, Z. Larssoniella duplicati n.sp. (Microsporidia, Unikaryonidae), a newly described pathogen infecting the double-spined spruce bark beetle, Ips duplicatus (Coleoptera, Scolytidae) in the Czech Republic. J. Pest Sci. 2006, 79, 127–135, doi: 10.1007/s10340-006-0124-y.
Line 44 can you please provide more information about climate change, because usually climate change in context of pest spreading is from south to north (many references about that)
- Added more information, lines: 45-51 and 52-60.
Lines 56-60 I think there are some information missing considering the specificity of L.t. infection on just one bark beetle species. There is evidence when pathogen occur only in a genus of bark beetle species. This is not the same as Chytridiopsis typographi which is present in many genera. I think this is worth to mention because the are some examples in the literature (Caningia in Tomicus or Pityokteines) and C.t. is also a part of the presented study.
- references were added (line 65):
- Pernek, M.; Matoševič, D.; Hrašovec, B,; Kučinić, M.; Wegensteiner, R. Occurrence of pathogens in outbreak populations of Pityokteines spp. (Coleoptera, Curculionidae, Scolytinae) in silver fir forests J. Pest Sci. 2009, 82, 343.
- Goertz, D.; Pernek, M.; Händel, U.; Kohlmayr, B.; Wegensteiner, R. Infection, course of disease and effects of Canningia tomici in Tomicus piniperda and Tomicus minor (Coleoptera: Curculionidae). Period Biol. 2017, 119, 285-293.
Lines 80-82 Why did you change the sites of traps from year to year (more or less)? Data would be more clear if just the same site would appear every year
- Yes, i tis true, but… It is not really easy to gain representative samples from consecutive years from the same study sites, because the sites and forest managers and forests themselves are changing hand in hand with the time. Therefore, we worked with samples from such wide range of countries, which we were able to get from our colleagues and were able to gain and transport to the Czech Republic for analysis. For possible next research it would be clearly better if the data could be collected from same study sites in few consecutive years…
Line 87 Can you provide exact how old was a specific forest on specific site? This possible would have an impact on the abundance and possible also on the infection level. This should be clarified
- clarified: lines: 94-95: All forest stands at study sites were composed of a mosaic of trees of all ages, so there was enough suitable material for Ips duplicatus infestation.
Line 140 It looks correct mathematically but are the data presented for CZ really correct' Because there is a high infection rate in just one site (?). Please give an explanation for that.
- Thank you for this note. This is reason why we do not used only mean, but also standard error of the mean.
Line 221-228 in line 208 you stated that infection of L.d. do not change with host population density but in this line it becomes unclear. Maybe you should rewrite these two lines making it more clear. They are some references where population of bark beetle in outbreaks an impact on infection levels of a pathogen have (eg Canningia Pityokteines) which should be mentioned too.
- True statement, thank you. We rewrite completely this paragraph and make it clearer. Lines: 218-227.
- Reference was added:
- Goertz, D.; Pernek, M.; Händel, U.; Kohlmayr, B.; Wegensteiner, R. Infection, course of disease and effects of Canningia tomici in Tomicus piniperda and Tomicus minor (Coleoptera: Curculionidae). Period Biol. 2017, 119, 285-293.

Reviewer 2 Report
My suggestions are contained in the text.

Author Response
Line 30 I suggest a Ips duplicatus instead of a the double-spined spruce bark beetle
- changed (line 29)
Line 42-43 but also for Central Europe and Siberia, which the authors write below
- The native ranges were put together on lines 40-42, and removed from line 52
Line 50 In 1931 Ips duplicatus shows Karpiński from the Bialowieza Forest Karpiński J.J. 1931. Korniki (Ipidae) Puszczy Białowieskiej. Pol. Pismo Ent., 10(1):18-39.
Area of spreading was specified, lines: 52-54

Round 2
Reviewer 1 Report
The authors made huge improvement after the review and I have no further question, so I recommended the manuscript for publication
Author Response
Thanks for your comments and notes